# Pre- and post-production processes increasingly dominate greenhouse gas emissions from agri-food systems

Francesco N. Tubiello[1], Kevin Karl[1,2], Alessandro Flammini[1,3], Johannes Gütschow[4], Griffiths Obli-Laryea[1], Giulia Conchedda[1], Xueyao Pan[1], Sally Yue Qi[2], Hörn Halldórudóttir Heiðarsdóttir[1], Nathan Wanner[1], Roberta Quadrelli[5], Leonardo Rocha Souza[6], Philippe Benoit[2], Matthew Hayek[7], David Sandalow[2], Erik Mencos-Contrera[8,9], Cynthia Rosenzweig[9,8], Jose' Rosero Moncayo[1], Piero Conforti[1] and Maximo Torero[1]

[1] Food and Agriculture Organization, Rome, Italy

[2] Center on Global Energy Policy, Columbia University, New York, USA

[3] United Nations Industrial Development Organization, Department of Environment, Vienna, Austria

[4] Potsdam Institute for Global Climate Research, Potsdam, Germany

[5] International Energy Agency, Paris, Franc

[6] United Nations Statistics Division, New York, USA

[7] Department of Environmental Science, New York University, New York, USA

[8] Center of Global Climate Research, Columbia University, New York, USA

[9] NASA Goddard Institute for Space Studies, New York, USA

*Corresponding author:* Francesco N. Tubiello, francesco.tubiello@fao.org

**Abstract.** We present results from the FAOSTAT *Emissions shares* database, covering emissions from agri-food systems and their shares to total anthropogenic emissions for 196 countries and 40 territories, for the period 1990-2019. We find that in 2019, global agri-food systems emissions were 16.5 (95% CI range: 11-22) billion metric tonnes (Gt $CO_{2eq}$ $yr^{-1}$), corresponding to 31% (range: 19-43%) of total anthropogenic emissions. Of the agri-food systems total, global emissions within the farm gate— from crop and livestock production processes including on-farm energy use— were 7.2 Gt $CO_{2eq}$ $yr^{-1}$; emissions from land use change, due to deforestation and peatland degradation, were 3.5 Gt $CO_{2eq}$ $yr^{-1}$; and emissions from pre- and post-production processes— manufacturing of fertilizers, food processing, packaging, transport, retail, household consumption and food waste disposal— were 5.8 Gt $CO_{2eq}$ $yr^{-1}$. Over the study period 1990-2019, agri-food systems emissions increased in total by 17%, largely driven by a doubling of emissions from pre- and post-production processes. Conversely, the FAOSTAT data show that since 1990 land use emissions decreased by 25%, while emissions within the farm gate increased 9%. In 2019, in terms of individual greenhouse gases (GHGs), pre- and post- production processes emitted the most $CO_2$ (3.9 Gt $CO_2$ $yr^{-1}$), preceding land use change (3.3 Gt $CO_2$ $yr^{-1}$) and farm-gate (1.2 Gt $CO_2$ $yr^{-1}$) emissions. Conversely, farm-gate activities were by far the major emitter of methane (140 Mt $CH_4$ $yr^{-1}$) and of nitrous oxide (7.8 Mt $N_2O$ $yr^{-1}$). Pre-and post-production processes were also significant emitters of methane (49 Mt $CH_4$ $yr^{-1}$), mostly generated from the decay of solid food waste in landfills and open-dumps. One key trend over the 30-year period since 1990 highlighted by our analysis is the increasingly important role of food-related emissions generated outside of agricultural land, in pre- and post-production processes along the agri-food system, at global, regional

and national scales. In fact, our data show that by 2019, pre- and post-production processes had overtaken farm-

gate processes to become the largest GHG component of agri-food systems emissions in Annex I parties (2.2 Gt

$CO_{2eq}$ yr$^{-1}$). They also more than doubled in non-Annex I parties (to 3.5 Gt $CO_{2eq}$ yr$^{-1}$), becoming larger than

emissions from land-use change. By 2019 food supply chains had become the largest agri-food system component

in China (1100 Mt $CO_{2eq}$ yr$^{-1}$); USA (700 Mt $CO_{2eq}$ yr$^{-1}$) and EU-27 (600 Mt $CO_{2eq}$ yr$^{-1}$). This has important

repercussions for food-relevant national mitigation strategies, considering that until recently these have focused

mainly on reductions of non-$CO_2$ gases within the farm gate and on $CO_2$ mitigation from land use change. The

information used in this work is available as open data with DOI 10.5281/zenodo.5615082 (Tubiello et al., 2021d).

It is also available to users via the FAOSTAT database (FAO, 2021a), with annual updates.

**Keywords:** Agri-food systems, GHG emissions, farm gate, land use change, supply chains

## 1. Introduction

Agriculture is a significant contributor to climate change as well as one of the economic sectors most at risk from it. Greenhouse gas (GHG) emissions generated within the farm gate by crop and livestock production and related land use change contribute about one-fifth to one-quarter of total emissions from all human activities, when measured in $CO_2$ equivalents (Mbow et al., 2019; Smith et al., 2014; Vermeulen et al., 2012). The impacts are even starker in terms of individual GHG emissions. Agriculture contribute nearly 50% of global anthropogenic methane ($CH_4$) and 75% of the total nitrous oxide ($N_2O$) emissions (FAO, 2021b; Gütschow et al., 2021; Saunois, et al., 2020). Once pre- and post-production activities along agri-food systems supply chains are included, food and agriculture activities generate up to one-third of all anthropogenic emissions globally (Crippa et al., 2021a,b; Rosenzweig et al., 2020; Tubiello et al., 2021a). This larger food systems perspective expands the potential for designing GHG mitigation strategies across the entire food system, i.e., over and above the more traditional focus on agricultural production and land use management that is currently found within countries' Nationally Determined Contributions (FAO, 2019).

Significant progress has recently resulted in the development of novel databases with global coverage of country-level data on agri-food systems emissions (Crippa et al., 2021a,b; Tubiello et al., 2021a). Tubiello et al. (2021a), in particular, provided a mapping of emission categories of the Intergovernmental Panel on Climate Change (IPCC)—used by countries for reporting their national GHG inventories (NGHGI) to the United Nations Framework Convention on Climate Change (UNFCCC)—unto internationally accepted food and agriculture concepts that are more easily understood by farmers and planners in countries, including in Ministries of Agriculture. By providing a correspondence between IPCC and FAO terminology, we seek to help countries to more adequately capture important aspects of food and agriculture activities within existing climate reporting, so that they can better identify effective climate actions across their agri-food systems (Fig. 1, adapted from Tubiello et al., 2021a). Firstly, the correspondence mapping expands the IPCC "agriculture" definition to include, in addition to non-$CO_2$ emissions from the farm, also the $CO_2$ generated in drained peatlands on agricultural land (Conchedda and Tubiello, 2020; Drösler et al., 2014) and by energy use in farm operations (FAO, 2020b; Flammini et al., 2021; Sims and Flammini, 2014). Secondly, it usefully disaggregates the 'Land Use, land use change and forestry' (LULUCF) of IPCC (2003) by separating out the emissions directly linked to food and agriculture activities, such as those generated by deforestation (Curtis et al., 2020; Tubiello et al., 2021c) and peat fires (Prosperi et al., 2020), from carbon removals, which are largely associated to processes in managed forests rather than on agricultural land (Grassi et al., 2021).

We present herein and discuss results from the first agri-food systems emissions database in FAOSTAT. The new database covers, as in previous versions (Tubiello et al., 2013) agriculture production activities within the farm gate and associated land use and land use change emissions on agricultural land. Importantly, it also includes estimates of emissions from pre- and post-production processes along food supply chains, including: fertilizer manufacturing, energy use within the farm gate, food processing, domestic and international food transport, retail, packaging, household consumption and food systems waste disposal. The database provides emissions data for four main GHG gases/categories ($CO_2$, $CH_4$, $N_2O$ and fluorinated gases) and their combined $CO_2eq$ levels. Data are available by country, over the period 1990-2019, as well as by regional and other relevant aggregations.

Importantly, data are provided in both IPCC and FAO classifications, facilitating the identification of national mitigation strategies across agri-food systems in countries, regionally and globally.

## 2. Materials and methods

Recent work (Rosenzweig et al., 2021; Tubiello et al., 2021a) helped to characterize agri-food systems emissions into three components: 1) Farm Gate; 2) Land Use Change; and 3) Pre- and Post-Production. Emissions estimates from the first two—generated by crop and livestock production activities within the farm gate and by the conversion of natural ecosystems to agriculture, such as deforestation and peatland degradation—are well established (IPCC, 2019). In particular, FAO disseminates annual updates in FAOSTAT (FAO, 2021; Tubiello, 2019). This paper expands the available FAOSTAT data to include estimates of emissions from pre- and post-production processes, s,, including energy use in fertilizer manufacturing; food processing; packaging; transport; retail; household consumption; and waste disposal.

### 2.1 Mapping Agri-food Systems Components

The new FAOSTAT data are provided, for each country, in both IPCC and FAO classifications. Specifically, on the one hand, data can be downloaded using IPCC emissions categories: *Energy*; *Industrial Processes and Product Use* (IPPU, henceforth referred to as Industry); *Waste*; *Agriculture*; *Land Use, Land Use Change and Forestry* (LULUCF); and *Other*. Both the total emissions from IPCC sectors are provided, as well as the portion directly related to agri-food systems. On the other hand, through the IPCC to FAO mapping discussed above and extending previous work (Tubiello, 2021a), data can also be downloaded in relevant FAO categories, covering emissions from: *Farm Gate*, *Land Use Change*; and *Pre- and Post-Production* processes (Fig. 1).

The FAOSTAT emissions estimates follow the IPCC (2006) "territorial approach," i.e., they are assigned to the countries where they occur, independently of production or consumption considerations. For example, $CO_2$ emissions from energy use in fertilizers manufacturing are accounted for in the producing country, while the $N_2O$ emissions from fertilizer used on a country's agricultural land for crop production are accounted for in that country. Similarly, emissions from energy use in agri-food systems activities are accounted for in countries where fuel combustion for that particular activity occurs, including electricity generation. The methods applied herein do not cover additional, upstream emissions associated with fuel supply chains, which are therefore not assigned to agri-food systems. More details on the scope of this work are found in section 2.3.

### 2.2 Emissions Estimates

FAO regularly disseminates emissions data for fifteen sub-domains in relation to the farm gate and land use change components of agri-food systems emissions, with published methodologies and results (i.e., Tubiello et al., 2021a). This manuscript relies in addition on new methods for computing emissions from pre- and post-production processes. Specifically, methods for emissions from energy use in fertilizers manufacturing, food processing, retail and household consumption, as well as refrigeration in retail are presented in Tubiello et al, (2021b), while Karl and Tubiello (2021 a,b) presented methods for estimating agri-food systems emissions in transport and waste disposal. Finally, emissions from on-farm energy use were developed by Flammini *et al.,* 2021). We refer the interested reader to those original publications for full details, while for completeness we also provide a sufficiently detailed summary of methods and coefficients as Supplementary Material ot this manuscript.

More generally, a step-wise approach was followed for the estimation of agri-food systems emissions, as follows

*Step 1*: identify, for each food systems component, the relevant international statistics needed to characterize country-level activity data (AD);

*Step 2*: determine the food-related shares of the activity data ($AD_{food}$) and assigns relevant GHG emission factors (EF) to each activity;

*Step 3*: implement the generic IPCC method for estimating GHG emissions ($E_{food}$), using inputs of activity data and emission factors from the first two steps, as follows:

$$E_{food} = EF*AD_{food} \tag{1}$$

*Step 4:* Impute missing agri-food systems GHG emissions data by component. This step was limited to pre- and post-production processes, and applied where country-specific activity data were lacking. The imputation method used PRIMAP, a complete dataset of emissions estimates for all IPCC sectors, by country, covering the period 1990-2019 (Gütschow et al., 2021). The PRIMAP dataset is already available in FAOSTAT for the computation of emissions shares of agriculture to the total anthropogenic total (FAO, 2019; Tubiello et al., 2021a). It compiles all available information on GHG emissions by country, including from official reporting. It was used internationally as the basis for an early, first-order estimate of agri-food systems shares in total GHG emissions (IPCC, 2019). Additionally, it was recently used in a UNFCCC Synthesis Report (UNFCCC, 2021) to assess world GHG emissions from all sectors in preparation of a stock take exercise that will be undertaken in 2022-2023 to assess countries' performance against their mitigation commitments under the Paris Agreement. The imputations in equation (1) were performed by applying to the PRIMAP sectoral emissions country-specific food system emissions shares (Tubiello et al., 2021b for more details).

**2.3 Global Warming Potentials Used**

The estimated emissions data expressed in $CH_4$ and $N_2O$ gases were converted to $CO_2$-equivalents by using the 100-year global warming potentials (GWP) of the IPCC (2014) Fifth Assessment Report, and specifically:

GWP-$CH_4$ = 28; GWP-$N_2O$ = 265; GWP-Fgases = 5195. The value for F-gases was obtained as an average of several distinct products (Tubiello et al., 2021b).

**2.3  Data uncertainty and limitations**

**2.3.1 Boundaries**

The processes covered herein do not span all processes attributable to agri-food systems. In particular, the scope of this work does not include, by design, upstream GHG emissions in the fuel chain, such as petroleum refining, as well as methane leaks during extraction processes and piping. These are expected to be not negligible if considered. While emissions from such sources can be estimated using a fixed fuel chain coefficient for certain fuel supply chains (see Crippa et al., 2021a), the authors do not consider such sources to be within scope of this work. GHG emissions attributable to electricity generation are included in the scope of this work, which itself excludes upstream GHG emissions in the fuel chain used to generate electricity (Flammini et al., 2021; Tubiello et al., 2021b).

Conversely, emissions of fluorinated gases (f-gases) from household refrigeration and from climate-controlled transportation were not included for lack of available country-level data for disaggregated cold chain elements. However, one estimate suggests that the majority (over 60%) of global food-related F-gas emissions occur in the retail stage, which is accounted for here in this work (International Institute of Refrigeration, 2021). Emissions from pesticide manufacturing were also not included due to the paucity of information and methodologies for their estimation at country level, in contrast to advanced work in fertilizers manufacturing (Brentrup et al., 2016; Brentrup et al., 2018; IFS, 2019). Bellardy et al. (2008) estimated global emissions from pesticides manufacturing to be roughly 72 (range: 3-140) Mt CO2eq yr$^{-1}$, roughly 1-2% of the pre- and post-production total estimated in this work.

### 2.3.2 Uncertainty

Uncertainties in FAOSTAT farm gate and land use change emissions estimates have been characterized elsewhere, and computed in line with IPCC (2006) guidelines as ranging 30—70% across component processes. For the purpose of this analysis, we assigned uncertainties of 30% and 50% respectively to the farm gate and land use change components of the FAOSTAT agri-food systems emissions, in line with previous work (i..e., Tubiello et al., 2013; 2021b). The uncertainties in the estimates of pre- and post-production activities described herein are by contrast less documented. On the one hand, uncertainties in underlying energy activity data and emissions factors are typically lower than for the other two components, ranging 5-20% (Flammini et al., 2022). On the other, the relative novelty in estimating food systems shares for a range of activity data across many processes makes our estimates more uncertain, with heavy reliance on literature results from a subset of countries and regions that are necessarily extended to the rest of the world (Karl and Tubiello, 2021a). For this reason, we assigned an overall uncertainty of 30% to the pre- and post-production component. This is higher than the uncertainty of the underlying energy processes, but more in line with values used in recent work (Crippa et al., 2021a). As shown below, considering a roughly equal, one-third contribution of the three components and their assigned uncertainties, an overall uncertainty of 40% was estimated for the agri-food systems emissions totals, applicable to countries and regional aggregates.

The above uncertainties are meant only as first rough estimates, useful to determine tentative 95% confidence intervals for the overall agri-food system component of FAOSTAT emissions. Significantly more research is needed for further refinements in future studies, in particular on better characterizing sub-regional and regional activity data and emissions coefficients, given the diversity in agri-food system typology and their dependence on physical geography and national socio-economic drivers. These limitations nonetheless reflect the paucity of activity data available to describe agri-food systems components and their trends, globally and regionally. While knowledge and data exist for regions and countries such as the EU, USA China, and India, much remains to be done in terms of regional and country specific coverage.

### 2.3.3 Areas for Advancement

Work towards estimating agri-food systems emissions at the country level can be advanced in several ways. The present approach could be expanded on by including other country- and region-specific studies that estimate trends in energy consumption across a range of similar activities as proxies— whether or not they are distinctly related to food. Furthermore, other data sources could help explain and estimate variations in agri-food systems between countries, such as: GDP per capita, urbanization levels, proxies for infrastructure and industrial development, and

geographic and climate considerations. The development of a methodology to estimate emissions from pesticides could be explored, as it would help complement the understanding of emissions associated with chemical use in agriculture, in addition to fertilizers. Emissions from machinery manufacturing and from upstream GHG emissions in the fuel chain could also be added to further refine the analysis. This work could be further expanded by focusing on specific food commodities— requiring an additional focus on international trade and on supply and demand patterns (Dalin and Rodríguez-Iturbe, 2016). Such analysis would ultimately enable consumers to understand the full carbon footprint of particular commodities across global supply chains, which can facilitate GHG mitigation actions taken at the consumer level (Poore and Nemecek, 2018). Furthermore, it would be also useful to further investigate the increasing role of bioenergy and renewables as important mitigation opportunities in the food sector (Clark et al., 2020, JRC, 2015; Pablo-Romero et al., 2017; Wang, 2014).

**Data availability**

The GHG emission data presented herein cover the period 1990-2019, at the country level, with regional and global aggregates. They are available as open data, with DOI 10.5281/zenodo.5615082 (Tubiello et al., 2021d) and via the FAOSTAT emissions shares database (FAO, 2021a).

## 3 Results

### 3.1 Global trends

The FAOSTAT dataset considered in this study estimates in 2019 total anthropogenic emissions at 52 Gt $CO_2$eq $yr^{-1}$ without land use, land use change and forestry emissions (LULCUF), and 54 Gt $CO_2$eq $yr^{-1}$ with LULUCF— consistently with recent estimates (IPCC, 2019). We use the latter figure to compute emissions shares. In 2019 world-total agri-food systems emissions, expressed in terms of 95% confidence intervals (CI) determined using an overall uncertainty of 40%, were 16.5 (CI range: 10-23) billion metric tonnes (Gt $CO_{2eq}$ $yr^{-1}$), corresponding to 31% (range: 19%-42%) of total anthropogenic emissions (Tab. 1). Of the food systems total, global emissions within the farm gate –from crop and livestock production processes including on-farm energy use—were 7.2 (range: 5-9) Gt $CO_{2eq}$ $yr^{-1}$; emissions from land use change, due to deforestation and peatland degradation, were 3.5 (range: 2-5) Gt $CO_{2eq}$ $yr^{-1}$; and emissions from pre- and post-production processes –manufacturing of fertilizers, food processing, packaging, transport, retail, household consumption and food waste disposal—were 5.8 (range: 4-8) Gt $CO_{2eq}$ $yr^{-1}$. Over the study period 1990-2019, agri-food systems emissions increased in total by 17%, though they have remained rather constant since about 2006 (Fig. 2). These trends were largely driven by a doubling of emissions from pre- and post-production processes, while land use emissions decreased by 25% and farm gate increased only 9%. In terms of single GHG, pre- and post- production processes emitted the most $CO_2$ (3.9 Gt $CO_2$ $yr^{-1}$) in 2019, preceding land use change (3.3 Gt $CO_2$ $yr^{-1}$) and farm-gate (1.2 Gt $CO_2$ $yr^{-1}$) emissions. Conversely, farm-gate activities were by far the major emitter of methane (140 Mt $CH_4$ $yr^{-1}$) and of nitrous oxide (7.8 Mt $N_2O$ $yr^{-1}$). Pre-and post- processes were also significant emitters of methane (49 Mt $CH_4$ $yr^{-1}$), mostly generated from the decay of solid food waste in landfills and open-dumps.

Emissions from within the farm gate and those due to related land use processes, including details of their sub-components, have been discussed in Tubiello et al. (2021a) and are regularly presented within FAOSTAT statistical briefs (e.g., FAO, 2020a; 2021b). Here we provide a detailed discussion of the components of agri-food systems emissions from pre- and post-production activities along supply chains and their relative contribution to the food

system totals (Fig. 3). Considering that the uncertainties used above are rough estimates, we will not report
uncertainties in the following analysis. Our data show that in 2019 emissions from deforestation were the single
largest emission component of agri-food systems, at 3.1 Gt $CO_2$ yr$^{-1}$, having decreased 30% since 1990. The second
most important component were non-$CO_2$ emissions from enteric fermentation (2.8 Gt $CO_2$eq yr$^{-1}$), with increases
of 13%. These were followed by emissions from livestock manure (1.3 Gt $CO_2$eq yr$^{-1}$) and several pre- and post-
production emissions, including $CO_2$ from household consumption (1.3 Gt $CO_2$eq yr$^{-1}$), $CH_4$ from food waste
disposal (1.3 Gt $CO_2$eq yr$^{-1}$), mostly $CO_2$ from fossil-fuel combustion for on-farm energy use (1.0 Gt $CO_2$eq yr$^{-1}$),
and $CO_2$ and F-gases emissions from food retail (0.9 Gt $CO_2$eq yr$^{-1}$). Importantly, our data show that growth in
pre- and post-production components was particularly strong, with emissions from retail increasing from 1990 to
2019 by more than seven-fold, while emissions from household consumption more than doubled over the same
period.

Finally, while emissions from agri-food systems increased globally by 16 percent between 1990 and 2019, their
share in total emissions decreased, from 40 percent to 31 percent, as did the per capita emissions, from 2.7 to
2.1 tonnes $CO_2$eq per capita (Fig 2.)

**3.2 Regional Trends**

Our results indicate significant regional variation in terms of the composition of agri-food systems emissions by
component (Fig. 4). Specifically, in terms of total agri-food systems emissions in 2019, Asia had the largest
contribution, at 7 Gt $CO_2$eq yr$^{-1}$, followed by Africa (2.7 Gt $CO_2$eq yr$^{-1}$), South America (2.4 Gt $CO_2$eq yr$^{-1}$) and
Europe (2.1 Gt $CO_2$eq yr$^{-1}$). North America (1.5 Gt $CO_2$eq yr$^{-1}$) and Oceania (0.3 Gt $CO_2$eq yr$^{-1}$) were the smallest
emitters among regions (Fig. 4). Focusing on GHG emissions beyond agricultural land, pre- and post-production
emissions in 2019 were largest in Asia (2.9 Gt $CO_2$eq yr$^{-1}$), followed by Europe and North America (0.8-1.1 Gt
$CO_2$eq yr$^{-1}$). Regions also varied in terms of how agri-food systems components contributed to the total (Tab. 2).
In 2019, pre- and post- production emissions were the largest food systems contributor in Europe (55%), North
America (52%) and Asia (42%). Conversely, they were smallest in Oceania (23%), Africa (14%) and South
America (12%). Additionally, the contribution of pre- and post-production processes along food supply chains
significantly increased since 1990, when in no region they were the dominant emissions component. Since then,
they doubled in all regions except in Africa—where it remained below 15%.

The data show which pre- and post-production process was most important by region (Tab. 2). In 2019, food
household consumption was the dominant process outside of agricultural land emissions in Asia (0.9 Gt $CO_2$eq yr$^{-}$
$^1$) and Africa (0.2 Gt $CO_2$eq yr$^{-1}$). Conversely, Europe, Oceania and North America pre- and post-production
processes were led by emissions from food retail (0.3-0.4 Gt $CO_2$eq yr$^{-1}$), while South America was dominated by
emissions from food waste disposal (0.2 Gt $CO_2$eq yr$^{-1}$).

**3.3 Country Trends**

Our estimates show a marked variation among countries in terms of total emissions as well as the composition of
contributions across farm gate, land use change and pre- and post-processing components (Fig. 5). China had the
most emissions (1.9 Gt $CO_2$eq yr$^{-1}$), followed by India, Brazil, Indonesia and the USA (1.2-1.3 Gt $CO_2$eq yr$^{-1}$).
Democratic Republic of Congo (DRC) and Russian Federation followed with 0.5-0.6 Gt $CO_2$eq yr$^{-1}$, followed by
Pakistan, Canada and Mexico with 0.2-0.3 Gt $CO_2$eq yr$^{-1}$. The contribution of the three main agri-food systems

components to the national total differed among countries significantly (Fig. 5). For instance, China and India had virtually no contribution from land use change to agri-food systems emissions. The land use contribution was also minor in the USA, Russian Federation and Pakistan. Conversely, the latter was the dominant emissions component in Brazil, Indonesia and the DRC. Additionally, the new database allowed for an in-depth analysis by country of pre- and post-production emissions along the agri-food chain, highlighting a significant variety in most relevant sub-process contribution (Tab. 3). For the year 2019, pre- and post-production emissions were dominated in China by food household consumption processes (463 Mt $CO_2$eq yr$^{-1}$), whereas food waste disposal was the dominant pathway in Brazil, Indonesia (77 Mt $CO_2$eq yr$^{-1}$), DRC (8 Mt $CO_2$eq yr$^{-1}$), Pakistan (33 Mt $CO_2$eq yr$^{-1}$) and Mexico, (56 Mt $CO_2$eq yr$^{-1}$). Emissions from food retail dominated the pre- and post-production component in the USA (292 Mt $CO_2$eq yr$^{-1}$), Russian Federation (177 Mt $CO_2$eq yr$^{-1}$) and Canada (20 Mt $CO_2$eq yr$^{-1}$). Finally, on-farm energy use was the largest pre- and post-production component in India (205 Mt $CO_2$eq yr$^{-1}$).

## 4 Discussion

### 4.1 Comparisons with previous work

The overall assessment of total agri-food systems emissions found in this work confirms recent previous findings by the IPCC (2019) and Crippa et al. (2021). With regards to pre- and post-production, the FAOSTAT estimates were consistent (Tab. 4) with previous findings (i.e., Crippa et al., 2021a, b; Vermuelen et al., 2012; Poore and Nemecek, 2018). In particular, emissions estimates for food transport, processing, waste and retail were consistent with EDGAR-FOOD (Karl and Tubiello, 2021b) and estimates for fertilizers manufacturing were in line with previous work by Vermeulen (2012). Conversely, FAOSTAT estimates were higher than EDGAR-FOOD for household consumption and lower for food packaging, the latter possibly linked to FAOSTAT estimates excluding indirect emissions from fuel supply chains, which were instead included in previously published estimates. Finally, our estimates of F-gas emissions from retail agreed well with those published in EDGAR-FOOD.

The most important disagreement with previous work was observed in relation to household consumption emissions. FAOSTAT estimates in this work, 1.2 Gt $CO_2$eq, were nearly three times those of EDGAR-FOOD (with reference to 2015, the last year for which EDGAR data was available). While much more research is needed to refine estimates in this important agri-food systems component, our estimates were in fact well aligned with earlier FAO (2011) work (Figure 4), as well as more consistent with observed population growth, an important determinant of household consumption trends.

### 4.2 Trends

One notable trend over the 30-year period since 1990 is the increasingly important role of food-related emissions generated outside of agricultural land, in pre- and post-production processes along food supply chains, at all global, regional and national scales. Our data show that by 2019, pre- and post-production processes had overtaken farm-gate processes to become the largest GHG component of agri-food systems emissions in Annex I parties (2.2 Gt $CO_{2eq}$ yr$^{-1}$). While farm gate emissions still dominated food-systems processes in non-Annex I parties, emissions from pre- and post-production were closing the gap in 2019, surpassing land use change, and having doubled since 1990 to 3.5 Gt $CO_{2eq}$ yr$^{-1}$. By 2019, pre- and post-production processes had become the largest agri-food system component in China (1.1 Gt $CO_{2eq}$ yr$^{-1}$); USA (0.7 Gt $CO_{2eq}$ yr$^{-1}$) and EU-27 (0.6 Gt $CO_{2eq}$ yr$^{-1}$). This has important repercussions for food-relevant national mitigation strategies, such as those included in countries' NDCs,

considering that until recently these have focused mainly on reductions of non-$CO_2$ gases within the farm gate and on $CO_2$ mitigation from land use change (Hönle et al., 2019).

Importantly, the FAOSTAT database presented here allows for an estimation of the percentage share contribution of food systems emissions in total anthropogenic emissions, by country as well as at regional and global levels, over the period 1990-2019. A number of important issues can be highlighted to this end (Tab. 5 and Fig. 6). First, in terms of CO2eq, the share of world total agri-food systems emissions decreased from 40% in 1990 to 31% in 2019. Thus while it is important to note that one-third of all GHG emissions today are generated by agri-food systems, their shares in total emissions may continue to decrease in the near future. This decreasing trend was driven by trends in large regions, consistently with transformations in their agri-food systems and land use change patterns. For instance, in South America, the region with the highest food systems share over the entire study period (Fig. 6), food shares decreased from 96% in 1990 to 72% in 2019. In Africa, from 67% to 57%, in Asia from 49% to 24% and in Oceania from 57% to 39%. In contrast to these trends, our data suggested that in regions dominated by modern agri-food systems, such as Europe and North America, the overall share of agri-food systems emissions in fact increased from 1990 to 2019, specifically from 24% to 31% in Europe and from 17% to 21% in North America. Such increases could be explained by increases in absolute emissions from pre- and post-production activities (Tab. 5), re-enforced by concomitant emissions decreases in non-food sector, especially energy systems (Lamb et al., 2022). The noted increase in absolute emissions from pre- and post-production activities was in fact present in all regions, leading to increases in the relative contributions to agri-food systems of this component, except for Africa.

An analysis on agri-food systems impacts on total GHG emissions would not be complete without a focus on component gases in addition to quantities expressed in $CO_2$eq. The FAOSTAT data confirm the trends form 1990 to 2019 seen for total CO2eq emissions, with important features (Tab. 6). First, the impact of agri-food systems on world total $CO_2$ emissions was 21% in 2019 (down from 31% in 1990), a respectable share considering the importance of carbon dioxide in any effective long-term mitigation strategy. While most regions had contributions around this value, ranging 13%-23% for North America, Oceania, Europe and Asia, the $CO_2$ contribution of agri-food systems was highest in Africa (52%) and South America (70%), largely in relation to land use change emissions, still significant therein. Europe and North America were the only regions where the $CO_2$ share of agri-food systems actually increased from 1990 to 2019, confirming the growing weight of pre- and post-production processes, which typically involve fossil-fuel energy use and thus emissions of $CO_2$ gas through combustion. Second, the data highlight the significant contribution of agri-food systems to 2019 world total emissions of $CH_4$ (53%) and $N_2O$ (78%), also confirmed at regional levels (Tab. 6), linked to farm gate production processes (Tubiello, 2019).

Finally, the data highlight a very large increase in agri-food systems contributions of F-gas emissions, which went from near zero in 1990 to more than one-quarter of the world total in 2019 –with larger contributions in many regions. Such a marked increase is consistent with the growth in use of hydrofluorocarbons (HFCs) as refrigerants in the food retail and other sectors, following the banning of CFCs in 1990 (Hart et al., 2020; International Institute of Refrigeration, 2021; Tubiello et al., 2021b). Our findings are furthermore consistent with the strong growth in F-gas emissions reported in recent studies (Minx et al., 2021; Park et al., 2021).

An important aspect of the dataset presented in this study is its provision of information mapped across IPCC and FAO categories alike. Specific IPCC sectors include *Agriculture* and *Land use, land use change and forestry* (*LULUCF*). The IPCC further considers the *Agriculture, Forestry and Other Land Use* (*AFOLU*). While countries report their agriculture and food emissions to the UNFCCC within National GHG Inventories, our findings highlight the importance to expand that reporting to a fuller agri-food systems view, one that properly weights the contribution of food to the global economy. Indeed, our results show that agri-food systems emissions in 2019 were one-third of total anthropogenic emissions. This important picture does not emerge from NGHGI reporting aligned to IPCC categories, according to which for instance, *LULUCF* and *AFOLU* emissions contributed respectively 4% and 15% of the total.

## 5 Conclusions

This paper provided details of a new FAOSTAT database on GHG emissions along the entire agri-food systems chain, including crop and livestock production processes on the farm, land use change activities from the conversion of natural ecosystems to agricultural land, and processes along food supply chains, from input manufacturing to food processing, transport and retail, including household consumption and waste disposal.

The data are provided in open access mode to users worldwide and are available by country over the time period 1990-2019, with plans for annual updates. The major trends identified in this work help locate GHG emissions hotspots in agri-food systems at the country, regional and global level. This can inform the process of designing effective mitigation actions in food and agriculture. This work adds to knowledge on GHG emissions from agriculture and land use— generally well established in the literature— by adding critical information on emissions from a range of pre- and post-production processes. The new data highlight the increasingly important role that pre- and post-production processes along supply chains play in the overall GHG footprint of agri-food systems, globally and in most countries, providing new insights into food and agriculture development trends and future mitigation options.

The granularity of the dataset allows, for the first time, to highlight specific processes of importance in specific countries or group of countries with similar characteristics. The relevance of the information being provided cuts across several national and international priorities, specifically those aiming at achieving more productive and sustainable food systems, including in relation to climate change. To this end, the work presented herein completes a mapping of IPCC categories, used by countries for reporting to the climate convention, to food and agriculture categories that are more readily understandable by farmers and ministries of agriculture in countries. This helps better identify agri-food systems entry points within existing and future national determined contributions. Finally, the methodological work underlying these efforts complements and extends recent pioneering efforts by FAO and other groups in characterizing technical coefficients to enable quantifying the weight of agri-food systems within countries' emissions profiles. The next steps in such efforts would need the involvement of interested national and international experts in compiling a first set of coefficients for agri-food systems as a pratical 'agri-food systems annex' to the existing guidelines of the Intergovernmental Panel on Climate Change, providing guidance to countries on how to better characterize food and agriculture emissions within their national GHG inventories.

**6. Disclaimer**

The views expressed in this paper are the authors' only and do not necessarily reflect those of FAO, UNSD, UNIDO and IEA.

## 7. Acknowledgements

FAOSTAT is supported by the FAO regular budget, funded by its member countries. We acknowledge the efforts of national experts who provide the statistics on food and agriculture as well as on energy use that are at the basis of this effort. All authors contributed critically to the drafts and gave final approval for the publication. We are grateful for overall support by the Food Climate Partnership at Columbia University.

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

1   TABLES

| Activity | Category | 1990 | 2019 | Change |
|----------|----------|------|------|--------|
| Net Forest conversion | Land Use Change | 4,392 | 3,058 | -30% |
| Enteric Fermentation | Farm-Gate | 2,494 | 2,823 | 13% |
| Livestock Manure | Farm-Gate | 1,101 | 1,315 | 19% |
| Household Consumption | Pre- and Post-Production | 541 | 1,309 | 142% |
| Waste Disposal | Pre- and Post-Production | 984 | 1,278 | 30% |
| On-farm energy use | Farm-Gate | 757 | 1,021 | 35% |
| Food Retail | Pre- and Post-Production | 128 | 932 | 631% |
| Drained organic soils | Pre- and Post-Production | 736 | 833 | 13% |
| Rice Cultivation | Farm-Gate | 621 | 674 | 9% |
| Fires | Land Use Change | 558 | 654 | 17% |
| Synthetic Fertilizers | Farm-Gate | 422 | 601 | 42% |
| Food Transport | Pre- and Post-Production | 327 | 586 | 79% |
| Food Processing | Pre- and Post-Production | 421 | 510 | 21% |
| Fertilizers Manufacturing | Pre- and Post-Production | 152 | 408 | 168% |
| Food Packaging | Pre- and Post-Production | 166 | 310 | 87% |
| Crop Residues | Farm-Gate | 161 | 226 | 40% |
| Forestland | N/A | -3,391 | -2,571 | -24% |

5   **Table 1**. GHG emissions (Mt $CO_2$eq) by agri-food systems component for all processes considered in this work.

6   Data on forestland removals are provided for completeness of land-based emissions available in FAOSTAT.

7   Uncertainties (not shown) are estimated at 30% for farm gate and pre- and post-production components and at 50%

8   for land use change processes.

| Region | Farm Gate | LUC | PPP | Total | %PPP | %PPP (1990) | Highest PPP | note |
|---|---|---|---|---|---|---|---|---|
| *Asia* | 3.2 | 0.9 | 2.9 | 7.0 | 42% | 24% | 0.9 | Household |
| *Africa* | 1.1 | 1.2 | 0.4 | 2.7 | 14% | 16% | 0.2 | Household |
| *South America* | 1.0 | 1.1 | 0.3 | 2.4 | 12% | 6% | 0.1 | Waste |
| *Europe* | 0.9 | 0.1 | 1.1 | 2.1 | 55% | 26% | 0.4 | Retail |
| *Northern America* | 0.6 | 0.2 | 0.8 | 1.5 | 52% | 35% | 0.3 | Retail |
| *Oceania* | 0.2 | 0.0 | 0.1 | 0.3 | 23% | 11% | 0.0 | Retail |

**Table 2**. Regional GHG emissions (Gt $CO_2$eq) by agri-food systems component, showing farm gate, land use

change (LUC), pre- and post-production processes (PPP) and total emissions Percentage contribution of PPP

shown for the year 1990 and 2019. The last two columns show the largest estimated contributing PPP activity by

region. Uncertainties are estimated to be 30% for farm gate and PPP activities, 50% for land use change.

| Country | Farm-gate | LUC | PPP | Total | Main PPP | Main PPP Name |
|---|---|---|---|---|---|---|
| **China** | 792 | 0 | 1102 | 1894 | 469 | Household Consumption |
| **India** | 768 | 0 | 618 | 1386 | 205 | On-farm |
| **Brazil** | 553 | 663 | 144 | 1360 | 79 | Waste Disposal |
| **Indonesia** | 491 | 658 | 132 | 1281 | 76 | Waste Disposal |
| **USA** | 477 | 60 | 696 | 1232 | 292 | Retail |
| **DRC** | 28 | 624 | 9 | 660 | 8 | Waste Disposal |
| **Russian Federation** | 146 | 35 | 362 | 542 | 177 | Retail |
| **Pakistan** | 205 | 7 | 71 | 283 | 33 | Waste Disposal |
| **Canada** | 97 | 96 | 81 | 274 | 20 | Retail |
| **Mexico** | 115 | 15 | 116 | 246 | 56 | Waste Disposal |

**Table 3**. Top ten country GHG emissions (Mt $CO_2$eq) by agri-food systems component and total food systems emissions, 2019. The last two columns show the dominant sub-component of pre- and post-production processes. Agri-food system GHG emissions from the top 10 countries represent 55% of global agri-food system emissions. Country level uncertainties those used for global and regional estimates.

| Food system component | FAO (2011)[1] | Vermeulen et al. (2012)[2] | Poore & Nemecek (2018)[3] | Ritchie (2019)[4] | Tubiello et al. (2021a)[5] | Crippa et al. (2021) EDGAR-FOOD[6] | This analysis[6] |
|---|---|---|---|---|---|---|---|
| Reference year | Mid-2000s | 2004–2007 | 2009–2011 | 2017 | 2019 | 2015 | 2019 |
| Fertilizer manufacturing | - | 0.3–0.6 | - | - | - | - | 0.4 |
| Food processing | 2.1 | 0.2 | 0.6 | 0.5 | 4.3 (incl. retail and household consumption) | 0.5 | 0.5 |
| Food packaging | | 0.4 | 0.6 | 0.7 | | 1.0 | 0.3 |
| Food transport | | | 0.8 | 0.8 | 0.5 | 0.9 | 0.6 |
| Food retail | | 0.7 | 0.4 | 0.4 | | 0.8 | 0.9 |
| Food household consumption | 1.2 | 0.2 | - | - | | 0.5 | 1.3 |
| Waste disposal | - | 0.1 | - | - | 1.0 | 1.6 | 1.3 |
| On-farm electricity generation | - | - | - | - | - | - | 0.5 |
| **TOTAL** | **3.3** | **1.9–2.2** | **2.4** | **2. 4** | **5.8** | **5.3** | **5.8** |

[1] Includes emissions from indirect energy inputs (e.g. manufacturing of machinery). Global estimate based on literature.

[2] Global estimate based on Chinese and British emission patterns and literature.

[3] Meta-analysis of life-cycle assessments

[4] Global estimate based on literature

[5] Global estimate largely based on country-level (bottom-up) analysis (relying on FAOSTAT and EDGAR-FOOD)

[6] Global estimate largely based on country-level (bottom-up) analysis

**Table 4**. Overview of pre- and post-food production GHG emission estimates from selected studies, Gt CO2eq. Adapted from Tubiello et al. (2021b).

| | Farm gate | | Land Use Change | | Pre- and Post-Production | | Agri-food Total | Systems |
|---|---|---|---|---|---|---|---|---|
| | **1990** | **2019** | **1990** | **2019** | **1990** | **2019** | **1990** | **2019** |
| **Africa** | 705 | 1139 | 1017 | 1220 | 323 | 388 | 2045 | 2747 |
| | 23% | 24% | 33% | 26% | 11% | 8% | 67% | 57% |
| **Asia** | 2595 | 3250 | 1273 | 865 | 1223 | 2930 | 5091 | 7044 |
| | 25% | 11% | 12% | 3% | 12% | 10% | 49% | 24% |
| **Europe** | 1603 | 854 | 88 | 83 | 589 | 1140 | 2280 | 2077 |
| | 16% | 13% | 1% | 1% | 6% | 17% | 23% | 31% |
| **North America** | 538 | 574 | 175 | 156 | 376 | 777 | 1089 | 1507 |
| | 8% | 8% | 3% | 2% | 6% | 11% | 17% | 21% |
| **South America** | 728 | 982 | 1974 | 1106 | 176 | 281 | 2878 | 2369 |
| | 23% | 30% | 64% | 34% | 6% | 9% | 93% | 72% |
| **Oceania** | 267 | 223 | 65 | 16 | 42 | 71 | 374 | 309 |
| | 40% | 28% | 10% | 2% | 6% | 9% | 57% | 39% |
| **World** | 6604 | 7214 | 4676 | 3503 | 2886 | 5827 | 14165 | 16544 |
| | 19% | 13% | 13% | 6% | 8% | 11% | 40% | 31% |

**Table 5**. Regional GHG emissions (Mt $CO_2$eq) by agri-food systems component and total food systems emissions, 2019. The last two columns show the dominant sub-component of pre- and post-production processes. Uncertainties (not shown) are estimated at 30% for farm gate and pre- and post-production components and at 50% for land use change processes.

| | 1990 | 2019 | 1990 | 2019 | 1990 | 2019 | 1990 | 2019 | 1990 | 2019 |
|---|---|---|---|---|---|---|---|---|---|---|
| | $CO_2eq$ | | $CO_2$ | | $CH_4$ | | $N_2O$ | | F-gases | |
| World | 40 | 31 | 31 | 21 | 60 | 53 | 79 | 78 | 0 | 27 |
| Africa | 67 | 57 | 65 | 52 | 63 | 58 | 90 | 87 | 0 | 20 |
| Northern America | 17 | 21 | 11 | 13 | 36 | 42 | 60 | 70 | 0 | 56 |
| South America | 93 | 72 | 97 | 70 | 82 | 75 | 94 | 92 | 0 | 6 |
| Asia | 49 | 24 | 38 | 16 | 66 | 49 | 84 | 80 | 0 | 9 |
| Europe | 23 | 31 | 13 | 23 | 46 | 47 | 70 | 74 | 0 | 28 |
| Oceania | 57 | 39 | 38 | 22 | 76 | 64 | 93 | 77 | 0 | 63 |

**Table 6**. World total and regional GHG agri-food systems emissions shares (%), 1990-2019, by single gas and

$CO_2eq$. Uncertainties in shares (not shown) are the same as those estimated for absolute emissions. See Crippa et

al. (2021a) for a specific list of HFCs used in agri-food systems, which form the basis of the F-gas emissions data

estimated in this work.

**FIGURE LEGENDS**

**Figure 1**. Mapping of emissions across agri-food systems. Left-hand panel: IPCC sectors and processes used in national GHG emissions inventories. Right-hand panel: food and agriculture sectors and categories aligned to FAO's definitions.

**Figure 2**. World-total GHG emissions from agri-food systems, 1990-2019. Color bars show contributions by emissions within the farm gate (yellow); land use change (green) and pre- and post- production along food supply chains (blue). Source: FAOSTAT (FAO, 2021). Also shown are emissions per capita (authors' own calculations).

**Figure 3**. World total 2019 GHG emission from agri-food systems, showing contributions on agricultural land (left panel) and from pre- and post- production along food supply chains (right panel). Net removals on forest land are also shown, for completeness. The sum of emissions from agricultural land and forest land correspond to the IPCC AFOLU category. Source: FAOSTAT (FAO, 2021).

**Figure 4**. Total GHG emission from agri-food systems by FAO regions, 2019. Color bars show contributions by emissions within the farm gate (yellow); land use change (green) and pre- and post- production along food supply chains (blue). Source: FAOSTAT (FAO, 2021).

**Figure 5**. Total GHG emission from agri-food systems by country, top ten emitters, 2019. Color bars show contributions by emissions within the farm gate (yellow); land use change (green) and pre- and post- production along food supply chains (blue). Source: FAOSTAT (FAO, 2021).

**Figure 6**. Top panel: Agri-food sytems emissions ($GtCO_2eq \ yr^{-1}$); Bottom panel: shares of agri-food systems in total anthropogenic emissions (%). Data shown by region, 1990-2019. Color bars show contributions component: farm gate (yellow); land use change (green) and pre- and post- production along food supply chains (blue). Source: FAOSTAT (FAO, 2021).

| IPCC | | Food Systems Activity | GHG | | | FAO | | |
|---|---|---|---|---|---|---|---|---|
| | | | CH$_4$ | N$_2$O | CO$_2$ | | | |
| AFOLU | LULUCF | Net Forest Conversion | x | x | x | LAND USE CHANGE | AGRICULTURAL LAND | FOOD SYSTEMS |
| AFOLU | LULUCF | Tropical Forest Fires | x | x | x | LAND USE CHANGE | AGRICULTURAL LAND | FOOD SYSTEMS |
| AFOLU | LULUCF | Peat Fires | x | | x | LAND USE CHANGE | AGRICULTURAL LAND | FOOD SYSTEMS |
| AFOLU | LULUCF | Drained Organic Soils | x | | x | FARM GATE | AGRICULTURAL LAND | FOOD SYSTEMS |
| AFOLU | AGRICULTURE | Burning - Crop residues | x | x | | FARM GATE | AGRICULTURAL LAND | FOOD SYSTEMS |
| AFOLU | AGRICULTURE | Burning - Savanna | x | x | | FARM GATE | AGRICULTURAL LAND | FOOD SYSTEMS |
| AFOLU | AGRICULTURE | Crop Residues | | x | | FARM GATE | AGRICULTURAL LAND | FOOD SYSTEMS |
| AFOLU | AGRICULTURE | Drained Organic Soils | | x | | FARM GATE | AGRICULTURAL LAND | FOOD SYSTEMS |
| AFOLU | AGRICULTURE | Enteric Fermentation | x | | | FARM GATE | AGRICULTURAL LAND | FOOD SYSTEMS |
| AFOLU | AGRICULTURE | Manure Management | x | x | | FARM GATE | AGRICULTURAL LAND | FOOD SYSTEMS |
| AFOLU | AGRICULTURE | Manure Applied to Soils | | x | | FARM GATE | AGRICULTURAL LAND | FOOD SYSTEMS |
| AFOLU | AGRICULTURE | Manure Left on Pasture | | x | | FARM GATE | AGRICULTURAL LAND | FOOD SYSTEMS |
| AFOLU | AGRICULTURE | Rice Cultivation | x | | | FARM GATE | AGRICULTURAL LAND | FOOD SYSTEMS |
| AFOLU | AGRICULTURE | Synthetic Fertilizers | | x | | FARM GATE | AGRICULTURAL LAND | FOOD SYSTEMS |
| ENERGY | | On-farm Energy Use | x | x | x | FARM GATE | AGRICULTURAL LAND | FOOD SYSTEMS |
| ENERGY | | Transport | x | x | x | PRE AND POST PRODUCTION | | FOOD SYSTEMS |
| ENERGY | | Processing | x | x | x | PRE AND POST PRODUCTION | | FOOD SYSTEMS |
| ENERGY | | Packaging | x | x | x | PRE AND POST PRODUCTION | | FOOD SYSTEMS |
| ENERGY | | Fertilizer manufacturing | x | x | x | PRE AND POST PRODUCTION | | FOOD SYSTEMS |
| ENERGY | | Household consumption | x | x | x | PRE AND POST PRODUCTION | | FOOD SYSTEMS |
| ENERGY | | Retail –Energy Use | x | x | x | PRE AND POST PRODUCTION | | FOOD SYSTEMS |
| Industry | | Retail –Refrigeration | x | x | x | PRE AND POST PRODUCTION | | FOOD SYSTEMS |
| WASTE | | Solid Food Waste | x | | | PRE AND POST PRODUCTION | | FOOD SYSTEMS |
| WASTE | | Incineration | | | x | PRE AND POST PRODUCTION | | FOOD SYSTEMS |
| WASTE | | Industrial Wastewater | x | x | | PRE AND POST PRODUCTION | | FOOD SYSTEMS |
| WASTE | | Domestic Wastewater | x | x | | PRE AND POST PRODUCTION | | FOOD SYSTEMS |

**Figure 1. Mapping of emissions across agri-food systems.** Left-hand panel: IPCC sectors and processes used in national GHG emissions inventories. Right-hand panel: food and agriculture sectors and categories aligned to FAO's definitions

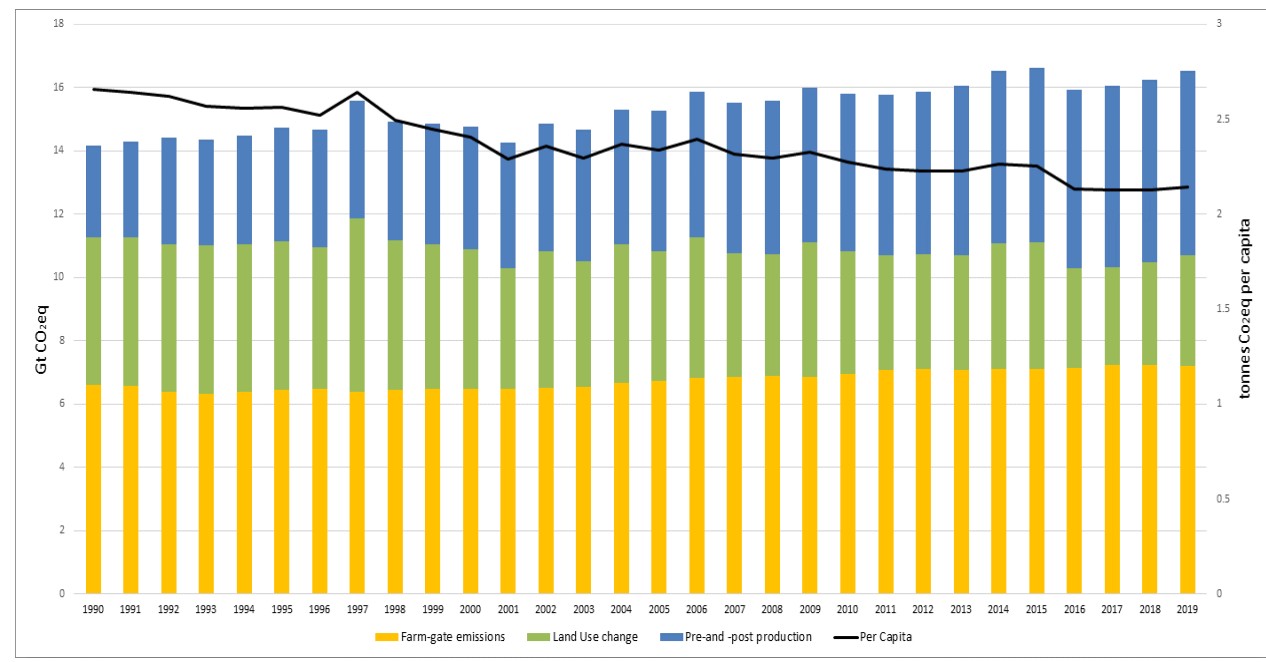

Figure 2. World-total GHG emissions from agri-food systems, 1990-2019. Color bars show contributions by emissions within the farm gate (yellow); land use change (green) and pre- and post- production along food supply chains (blue). Source: FAOSTAT (FAO, 2021). Also shown are emissions per capita (authors' own calculations).

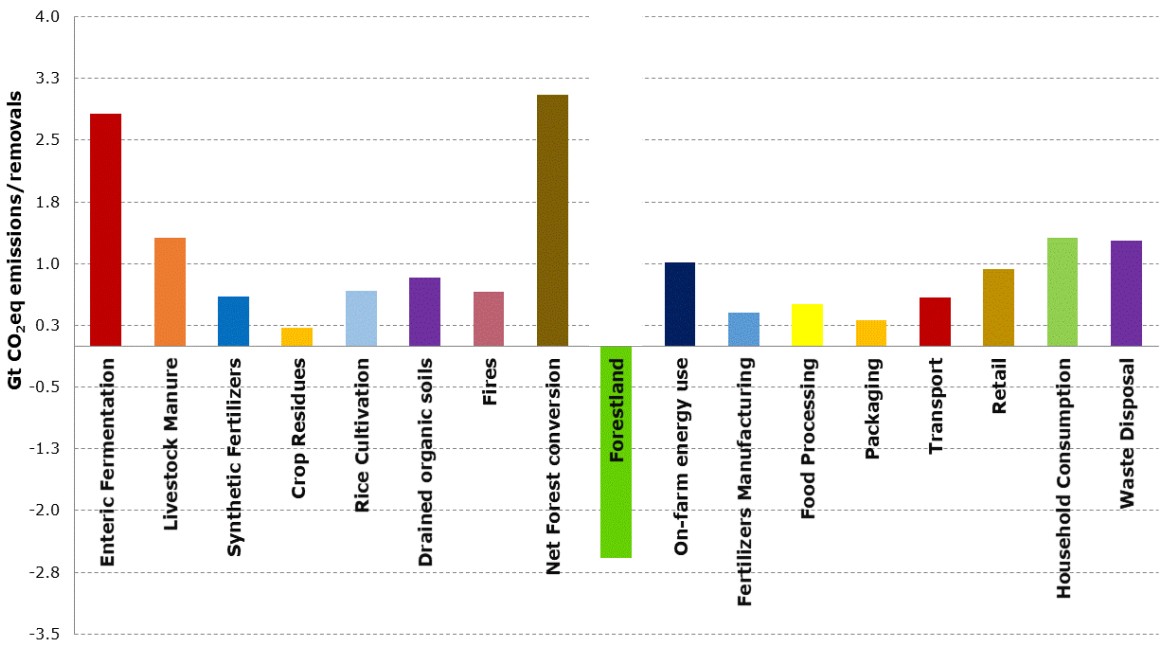

2 **Figure 3. World total 2019 GHG emission from agri-food systems.** Contributions on agricultural land are displayed on the
3 left, and from pre- and post- production along food supply chains on the right. Net removals on forest land are also shown in
4 the center for completeness. The sum of emissions from agricultural land and forest land correspond to the IPCC AFOLU
5 category. Source: FAOSTAT (FAO, 2021).

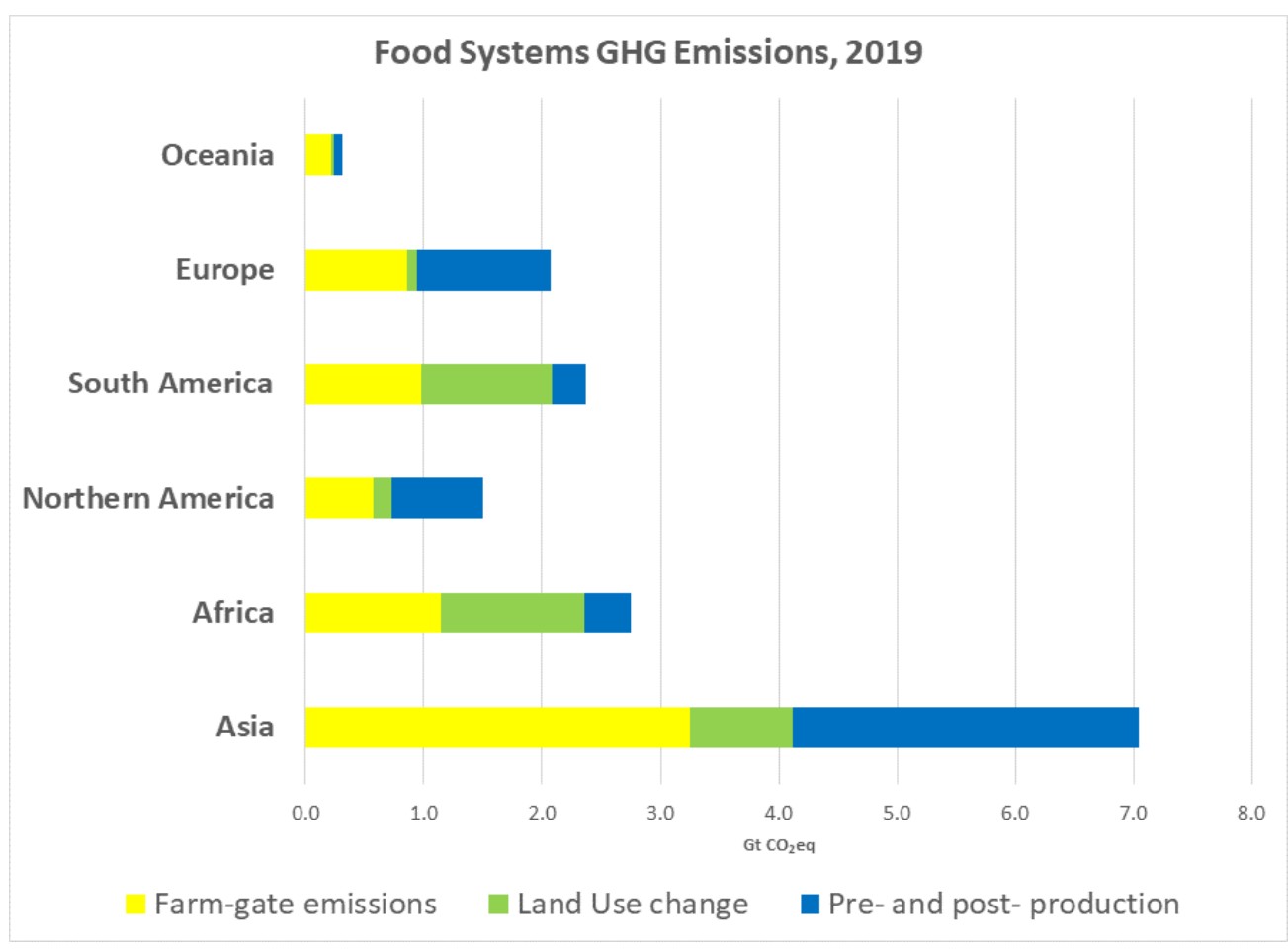

**Figure 4. Total GHG emission from agri-food systems by FAO regions, 2019.** Color bars show contributions by emissions within the farm gate (yellow); land use change (green) and pre- and post- production along food supply chains (blue). Source: FAOSTAT (FAO, 2021).

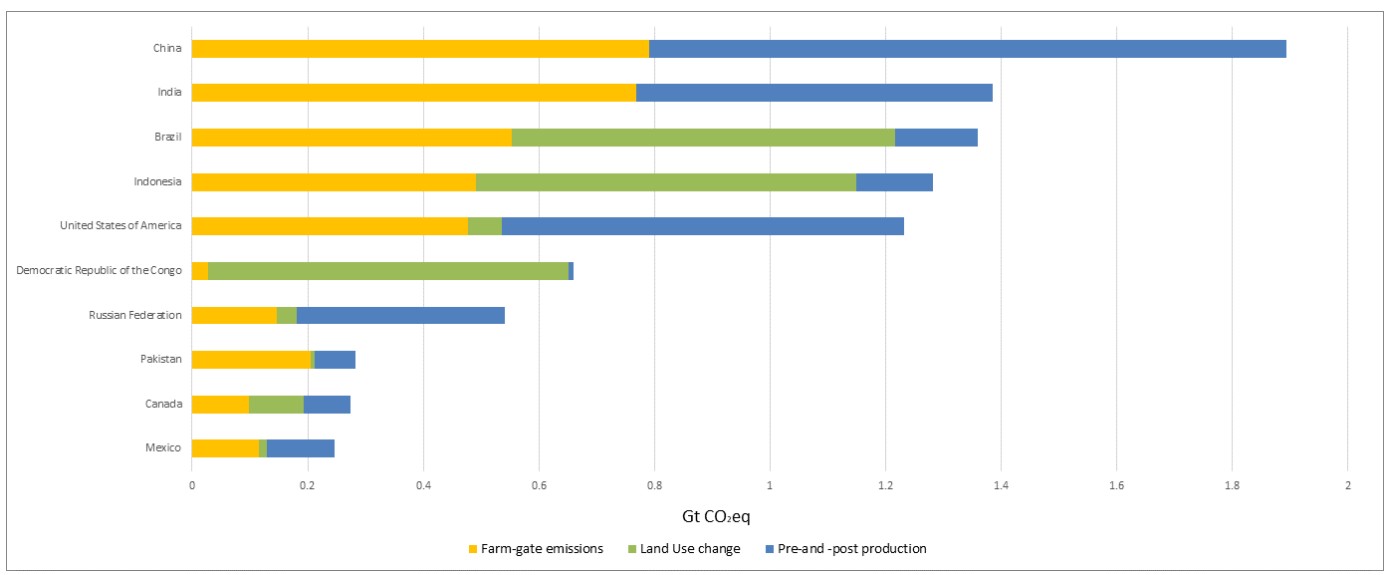

2 **Figure 5. Total GHG emission from agri-food systems by country, top ten emitters, 2019.** Color bars show contributions

3 by emissions within the farm gate (yellow); land use change (green) and pre- and post- production along food supply chains

4 (blue). Source: FAOSTAT (FAO, 2021).

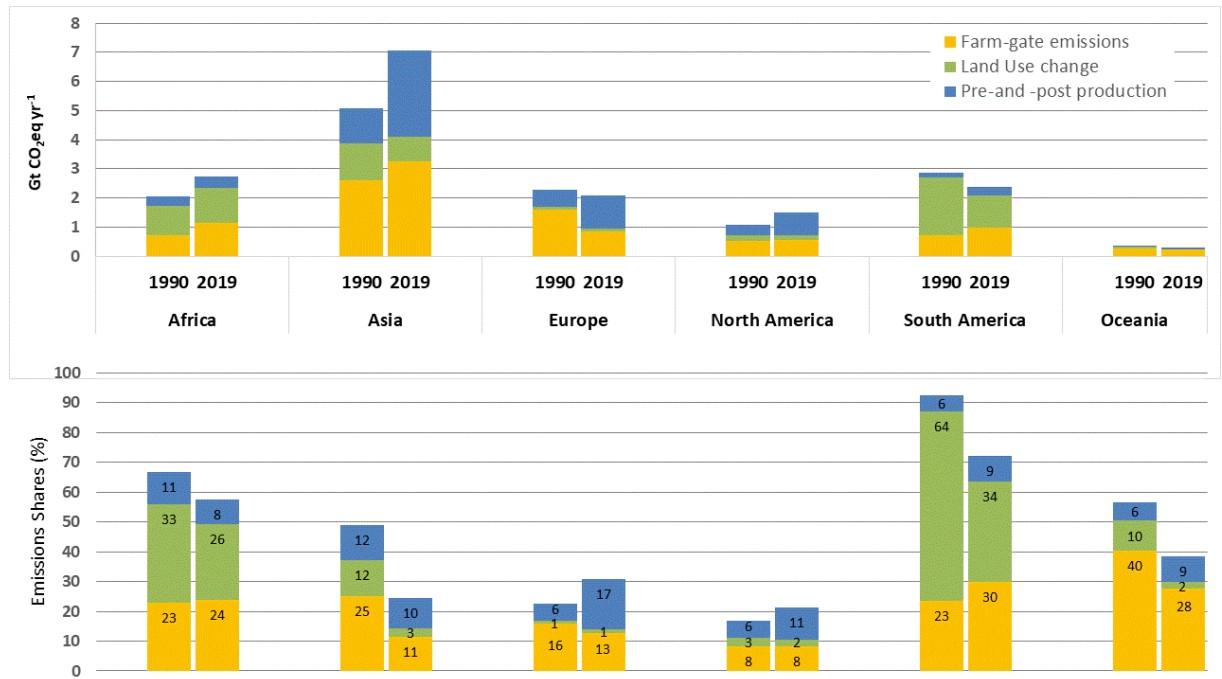

3 **Figure 6. Top panel: Agri-food systems emissions (GtCO$_2$eq yr$^{-1}$).** Bottom panel: shares of agri-food systems in total

4 anthropogenic emissions (%). Data shown by region, 1990-2019. Color bars show contributions component: farm gate (yellow);

5 land use change (green) and pre- and post- production along food supply chains (blue). Source: FAOSTAT (FAO, 2021).

