# Peer review of "Pre- and post-production processes increasingly dominate"

_Earth System Science Data, 2021_

## Author Comment (AC1)

**Author Responses to Referee Comments**

**RC1:** Page 1, Line 28: is it FAO or FAOSTAT (line 20)?

**Author Response**: We used FAOSTAT here for both.

**RC1:** Line 30: "in terms of single GHG" change to "in terms of individual greenhouse gases (GHGs)"

**Author Response:** This fix was made in text at line 30.

**RC1:** Line 34: the time period (1990-2019) is mentioned twice, at the beginning and end of the sentence

**Author Response:** Addressed in text in line 35.

**RC1:** Page 3. Line 2: typo in the first sentence, should read "as well as one of the economic sectors most at risk from it"

**Author Response:** Addressed in text in section 1.

**RC1:** Line 8: EDGAR-FOOD would be another important reference to include in this sentence (https://www.nature.com/articles/s43016-021-00225-9)

**Author Response:** We cited this reference in Section 1.

**RC1:** Page 4. Line 31: typo in 2022-2023

**Author Response:** Addressed in text in section 2.2.

**RC1:** Line 35- page 5 line 4: These sentences belong in the subsequent section on uncertainty.

**Author Response:** These lines were moved to the section on uncertainty (section 2.3.2)

**RC1:** Page 5. Line 5-11: Can the authors restructure to make a clear distinction between emissions sources that are (a) not included because they are indirect and out of scope ("upstream GHG emissions, refining, etc.") and (b) not included because data was not available, even though they are direct and within scope?

**Author Response:** Both of these issues are addressed in the revised section 2.3.1.

**RC1:** It would be important to note in (a) whether or not indirect emissions from electricity use are also excluded, as this is generally the largest indirect source across all sectors; and in (b) how significant these sources are in estimated CO2 equivalents, and whether this is a complete list of omitted direct emissions sources.

**Author Response:** Addressed in section 2.3.1. Estimates of significance presented for cold chain elements not covered are presented, as well as estimates for pesticide manufacturing. Fuel chain estimates not provided since they are not considered in scope.

**RC1:** Line 12-21: This is a relatively short discussion of uncertainty – given its importance in the context of food system emissions. As stated above, several sentences from the prior section could be brought down. Several further points could be made:

Does the estimated uncertainty range ("30—70% across many processes (Tubiello, 2019)") also hold true for this dataset? Please be explicit.

Could uncertainty estimates be provided for sub-components of the data (e.g. by gas, or food system component)? This is critical information for the data users.

To what extent does uncertainty prevent us from making policy relevant statements on (1) total emissions levels, (2) total emissions trends, (3) the relative importance and impact of different food system components?

Does uncertainty increase with decreasing scale (global to regional to country level data)?

**Author Response:**

Thank you for making such an important point. We have expanded the discussion on uncertainty to address the points raised, including presenting an uncertainty range for the high-level global results, by agri-food systems component, both in the abstract and in the results section. We feel this rough additional estimates make the presentation of our findings more interesting and useful to readers. Specifically, while recognizing that uncertainty estimation is difficult for most of the new data presented in this work, we have nonetheless used previous assessments of uncertainty in farm gate (30%) and land use processes (50%) and combined it with a first assessment of uncertainty in pre- and post-production processes (30%) mediated in part from the literature and based on our own assessment. At the same time, we recognize and make explicit in the revised text that these are preliminary uncertainty estimates, that need refinement in future work. To this end, but consistently with formulas for error propagation across linear sums, we assume scale-invariance from country to global and state so in several places within Table legends where numbers are presented.

Also because of the preliminary nature of this assessment, we feel that a discussion on the implications of uncertainty for policy relevant decision making is (of course well understood) out of scope in this manuscript.

**RC1:** Page 6. Line 7: Perhaps state the denominator here too (total global GHG emissions) and its source? It is also not in Table 1. (I see that it appears in the discussion. Please move up here.) You might consider placing it in the abstract too, since the sentence appears there too.

**Author Response:** Addressed in text at beginning of section 3.1.

**RC1:** Line 7: What would be the emissions range for the global total ($\pm$ xGtCO2eq yr-1), given the previously stated uncertainty?

**Author Response:**

See above. We have estimated uncertainties globally and by food system component, providing confidence intervals for the major high-level results in our database.

**RC1:** Page 8. Line 2-4: This is an important claim, also in the abstract. Can it be sourced? What is the measure of "national mitigation strategies"? Sector based targets within NDCs?

**Author Response:** Addressed in section 4.2.

**RC1:** Line 17-22: Presumably it is also due to shifts in other sectors, e.g. all else equal, reductions in power sector emissions will increase the proportion of food system emissions in the total. And power sector emissions have been declining in most EU countries and the US (e.g. https://www.tandfonline.com/doi/full/10.1080/14693062.2021.1990831)

**Author Response:** Addressed in section 4.2, second paragraph.

**RC1:** Line 37: The result on F-gases is surprising - and interesting. Can the authors provide a little more detail? Which are the main gases? Perhaps a link could be made to Minx et al. 2021, which corroborates F-gas growth in inventories with atmospheric inversions (Fig 2 https://essd.copernicus.org/articles/13/5213/2021/essd-13-5213-2021.html) Also, in Table 5, F-gases were 0 in 1990. Is this a data artefact? Or is it due to Montreal gases being replaced by HFCs/PFCs in the intervening decades?

**Author Response:** Addressed in section 4.2, fourth paragraph.

**RC1:** Page 9. Line 1-6: The language here suggests these subcomponents are trivial sources ("only", "mere"). Arguably 15% or even 3-4% is not trivial, so I would simply present the numbers without inferring their importance. If one wants to make a normative point, I would argue that all emissions sources should be considered important and worth policy attention.

**Author Response:** Good point, such language has been removed.

**RC1:** Line 12-32: There are multiple typos and phrasing errors here that could be improved. Please carefully check. Please also consider splitting this long paragraph into smaller chunks each with a substantive point.

**Author Response:** Addressed in text. The conclusion has been split into more paragraphs and rewritten for clarity.

**RC1:** Other comments on the manuscript: Table 1: Could headings be added to group these sources into their higher level categories, e.g. as in Figures 1 and 2?

**Author Response:** Addressed in table 1.

**RC1:** Table 3: You could add the fraction of global food system emissions that the top 10 add up to, in the caption.

**Author Response:** Placed in caption in table 3.

**RC1:** What global warming potentials are applied to estimate GHG emissions in CO2eq?

**Author Response:** Addressed in methods, section on Global Warming Potentials.

**RC1:** Comments on the dataset:

My first impression is that the dataset is too large (200mb), unstructured, and lacking important metadata. Together these make it only available for advanced users. Some specific comments:

If one opens the .csv in Excel, a warning comes up that the data is not fully loaded (too many rows). Could it be split into several files? Or could a basic user-friendly excel version be provided alongside the raw csv file, perhaps for a useful series of aggregates (e.g. global emissions by food system component, by gas, by region/country), or the full data just for high emitters/regions? Such simplified sheets would presumably be important to assist national agricultural ministries to better understand emissions along the supply chain (a claim in the manuscript).

There is no explanation of the column headings embedded in the file (What are the flags? What are the codes? Are two codes for years really needed?). For example, a basic user wouldn't know that Area contains both countries and regions, and Element contains two separate variables for five different gases (I would personally split this in two and have a gas column).

There are no country ISO codes, which raises barriers to joining other datasets (e.g. population, gdp).

Most tricky: what is the hierarchy and structure of the "Item" column? If I filter by "World", "2019", and "Emissions (CO2eq) (AR5)", the sum of Value is 228 GtCO2eq. So there is double counting among the Items. Which items do I exclude to produce the number in the manuscript – 16.5 GtCO2eq? I see already that "Energy" is included (37GtCO2eq) and shouldn't be. How do I know which items are in and which are out of the food system account? Could you add a column for this, so we don't have to use complicated string operations?

Can we have the GHGs in native units, so that different global warming potential metrics can be applied? (Or conversely, a column with the applied AR5 GWPs)?

**Citation**: https://doi.org/10.5194/essd-2021-389-RC1

**Author Response:** All above points have been addressed by providing a revised dataset and accompanying description in Zenodo. DOI has been updated accordingly.

**RC2:** The dataset is of interest but the methodology and underlying data is not described in the article. It is described in FAO Statistics Working Paper Series working papers, but it is not acceptable to have the methodology central in the data setting not described in the article (or in other peer reviewed articles). In particular, those methodologies are supposed to be peer reviewed, and also available (possibly as supplementary material) with a reviewed article. The methodologies from those working papers can be shortened, but upon reading them it seems that simply copying over most of the information, maybe with a summary in the main paper and a development in a supplementary material, or all in the main paper depending on the style of the review would be good as they are well written and describe adequately the methodologies. Another reason to bring those in the article is that there may be some additional peer review comments based on those methodologies.

It is somewhat unclear if additional data should be provided along with the main dataset. For instance shares of the food system. However this cannot really be discussed if the underlying methodology is not presented and discussed.

Most of the informations and the data corresponds to an already existing article, Tubiello et al., 2021a "Greenhouse gas emissions from food systems: building the evidence base". Therefore I am not sure about originality, but it may be normal as here the dataset is the focus. It makes all the more important to describe the methodology in the data article as it would be some originality.

**Author Response:**

We thank the reviewer for this comment. On the one hand, we feel strongly that methods underlying the development of a new database, for which structure, data, results and analysis are presented here, be separately peer-reviewed and published elsewhere. Too often authors hide the nitty gritty details of major methodological work in a SI section of a paper, with the result that often they are poorly understandable, as well as a burden on reviewers who suddenly need to split their tasks into a review of scientific findings given a method, plus a review of the actual new methods being proposed. This is why we provide the background methodology in three separate, peer-reviewed documents published by FAO. The important thing to highlight is that such papers are easily available online –so much so that more than one reviewer was able to access them in the course of this review process.

On the other hand, we recognize that there is no perfect formula to the above quandary, and that some sort of complete package is also of value. For this reason, we have revised the methods text to include an extended discussion of steps followed in the creation of the database and improved reference to the external methods paper. At the same time, we have made a robust summary of the methods developed externally and created a SI version that can be associated to this manuscript. Notwithstanding the fact that the original and more fully developed material can be found online as external papers. We trust that this solution is acceptable.

**RC2:** The dataset combines different and incompatible disaggregations and nomenclatures, which is an interesting and important point of the methodology. There is an explanation of the relationships between the nomenclatures in figure 1, and in the https://zenodo.org/record/5615082 page. It is badly explained in the article, only very briefly in 2.1, although describing the data should be important in the article.

For the general public, as the dataset combines different and incompatible disaggregations and nomenclatures it is not clear if it would be of interest. Although it is important to have those informations to understand the methodology and how these data can be derived from the PRIMAP data based on the IPCC nomenclature, for a non-specialist this makes a very unclear dataset.

**Author Response:** Addressed with new dataset in Zenodo and accompanying description therein and in the revised manuscript.

**RC2:** A comparison with Crippa et al would also be welcome as it is a similar work with care to explain what is exactly the ame when crippa et al has been used as a source. It is already done adequately, as far as I can tell from my readings in the Working Paper Series working papers, but it should be in the peer reviewed article and may trigger additional comments here.

**Author Response:** Addressed in new table inserted (Tab. 4.).

**RC2:** More remarks

p 4 l 33 and following, the discussion about uncertainty does not add much information, all the information is quite generic. There is some validation done in the FAO Statistics

Working Paper Series articles, theis should be presented/discussed here.

**Author Response:**

Thank you. We have revised the discussion on uncertainty within the manuscript, linking it concisely to findings in the methods papers in relation to pre- and post-production processes in particular. At the same time, we have added a SI document where some of the more specific discussion on uncertainty related to activity data and emissions factors is now also available within ESSD.

**RC2:**

p 4 l 25 The Step 4 of imputation of missing emissions is not clear (missing how?). It should be associated with additional data showing which data is imputed and which data is not.

**Author Response:** Addressed in section 2.2, second paragraph.

**RC2:** p 6 l 35 3.2 Regional Trends

The numbers per regional blocks or countries are not very interesting as the populations may be very different. Also some goods may be exported which makes these numbers also difficult to interpret. Some emissions are directly linked with the consumption, so should be local, but it is not the case for processing, packaging and fertilizer production.

**Author Response:** Global, Annex-I and Non-Annex-I per capita are provided in section 3.1, 3rd paragraph. Explanation of how GHGs are accounted for have been additionally included in section 2.1, second paragraph.

**RC2:** p 8 l 7 the database FAOSTAT-PRIMAP is not introduced before nor really presented. It should be presented and even be available with this data, as if I understand well it is the data which corresponds to the methodology, the data presented is an aggregation.

**Author Response:** Addressed in text.

**RC2:** A minor remark, since the data is about reorganizing disaggregated data in different categories, the comparison of nomenclatures can be of interest in term of methodology to understand the strength and limitations of each nomenclature and warn about uses. However, this is not done at all in the article.

**Citation**: https://doi.org/10.5194/essd-2021-389-RC2

**Author Response:** We believe this discussion is best left to other work focusing on the issue, which we have cited here.